# Assessment of Cassava Pollen Viability and Ovule Fertilizability under Red-Light, 6-Benzyl Adenine, and Silver Thiosulphate Treatments

**DOI:** 10.3390/plants13141988

**Published:** 2024-07-20

**Authors:** Julius K. Baguma, Settumba B. Mukasa, Mildred Ochwo-Ssemakula, Ephraim Nuwamanya, Paula Iragaba, Enoch Wembabazi, Michael Kanaabi, Peter T. Hyde, Tim L. Setter, Titus Alicai, Benard Yada, Williams Esuma, Yona Baguma, Robert S. Kawuki

**Affiliations:** 1School of Agricultural Sciences, Makerere University, Kampala P.O. Box 7062, Uganda; sbmukasa@gmail.com (S.B.M.); mildred.ochwossemakula@mak.ac.ug (M.O.-S.); nuwamanyaephraim@gmail.com (E.N.); 2National Crops Resources Research Institute (NaCRRI), Namulonge, Kampala P.O. Box 7084, Uganda; iragapaula@gmail.com (P.I.); earlmbabazi@gmail.com (E.W.); kanaabimichael@gmail.com (M.K.); talicai@hotmail.com (T.A.); yadabenard21@gmail.com (B.Y.); esumawilliams@yahoo.co.uk (W.E.); 3Soil and Crop Sciences, School of Integrative Plant Science, Cornell University, Ithaca, NY 14853, USA; pth7@cornell.edu (P.T.H.); tls1@cornell.edu (T.L.S.); 4National Agricultural Research Organization (NARO) Secretariat, Entebbe P.O. Box 295, Uganda; ybaguma@naro.go.ug; 5World Coffee Research (WCR), Portland, OR 97225, USA; kawukisezirobert@gmail.com

**Keywords:** pollen viability, in vitro stainability, in vivo germinability, ovule fertilizability, pollen diameter, pollen ploidy

## Abstract

Understanding pollen and ovule fertility as factors influencing fruit and seed set is important in cassava breeding. Extended daylength with red light (RL) and plant growth regulators (PGRs) have been used to induce flowering and fruit set in cassava without any reference to effects on pollen viability or ovule fertilizability. This study investigated the effects of field-applied RL and PGR on pollen viability and ovule fertilizability. Panels of cassava genotypes with early or moderate flowering responses were used. RL was administered from dusk to dawn. Two PGRs, 6-benzyl adenine (BA), a cytokinin and silver thiosulphate (STS), an anti-ethylene, were applied. Pollen viability was assessed based on pollen grain diameter, in vitro stainability, in vivo germinability, ovule fertilizability, and ploidy level. Treating flowers with RL increased the pollen diameter from 145.6 in control to 148.5 µm in RL, 78.5 to 93.0% in stainability, and 52.0 to 56.9% in ovule fertilizability in treated female flowers. The fruit set also increased from 51.5 in control to 71.8% in RL-treated female flowers. The seed set followed a similar trend. The ploidy level of pollen from RL-treated flowers increased slightly and was positively correlated with pollen diameter (*R*^2^ = 0.09 *), ovule fertilization (*R*^2^ = 0.20 *), fruit set (*R*^2^ = 0.59 *), and seed set (*R*^2^ = 0.60 *). Treating flowers with PGR did not affect pollen diameter but increased stainability from 78.5% in control to 82.1%, ovule fertilizability from 42.9 to 64.9%, and fruit set from 23.2 to 51.9% in PGR-treated female flowers. Combined BA + STS application caused the highest ovule fertilizability, fruit, and seed set efficiency. These results show that RL and PGR treatments increase pollen viability and ovule fertilizability. This is important for planning pollination strategies in cassava breeding programmes.

## 1. Introduction

Cassava (*Manihot esculenta* Crantz) is a staple crop with tuberous roots which are rich in starch, making it important as a food for humans and animals and as a raw material for different industries [1,2,3]. It is a crop resistant to unpredictable climatic fluctuations as well as poor soils [4,5]. Thus, it has potential both for bridging the food demand gap and as a drought-tolerant crop in the forecasted warmer 2030 world. This necessitates that breeding efforts need to prioritize the improvement of important traits such as yield- and nutrient-related traits, as well as tolerance to both abiotic and biotic constraints. However, in order to achieve these breeding targets, a deep understanding of pollen and ovule fertility as factors influencing fruit and seed set is needed.

One of the impediments to the genetic improvement of cassava is low or no seed set due to the high abortion rates of flowers or fruits [6]. High rates of abortions and low fruit or seed set rates in controlled crosses, averaging less than one seed per pollination, have been reported in cassava crossing programmes [7]. Indeed, among a diverse range of plant species, the abortion of a large proportion of flowers and fruits is an enormous challenge, with some species producing barely one fruit for every 1000 or more flowers produced [8].

Key factors underpinning fruit abortions in many plant species include parental genetic differences; unsuccessful pollinations due to poor quality and/or unviable pollen; low ovule fertility rates; inadequate, unsynchronized, or overproduced amounts; and the inhibitory effects of growth hormones [8,9]. In conventional plant breeding, successful pollination is a prerequisite for fertilization and seed set [10,11], and this highly depends on good-quality and viable pollen, as well as stigma receptivity [12]. Thus, knowledge of pollen viability enhancement or the identification of viable pollen, its germination ability, and it capacity to fertilize ovules is essential for increasing fruit and seed set in crop breeding programs [13]. However, knowledge of the factors influencing pollen viability and the success or failure of fruit and seed set in cassava is still scanty. It is worth noting that manipulation of male fertility has proved a useful trait for breeding and increased crop yield [14,15].

Viable pollen is important for species’ survival in the next plant generation, as well as in targeted breeding and crop improvement programmes [13]. Pollen viability, its functionality or germination, and its capacity to fertilize ovules depend on various factors, including enzyme-activity, cytoplasmic contents, nutrition status, plant growth regulators, varietal differences, size or ploidy level, and environmental factors, including photoperiodism [16,17,18]. The roles of plant growth regulators (PGR) and photoperiodism in plant physiology have been extensively studied. However, there are only a few studies on their effects on pollen viability. For example, PGRs like brassinosteroids have been used to increase pollen viability in pomegranate flowers [16] and *Tulipa greigii* [19].

On the other hand, the application of red and blue lights (in a ratio of 1:1) was used to increase pollen viability in tomatoes [20], and blue and yellow lights promoted pollen germination in *Peltophorum* [21]. In cassava, field applications of RL photoperiod extension and PGRs (BA and STS) have been reported to enhance fruit and seed set [22], but their possible effects on pollen viability and ovule fertilizability were not explored. And yet, pollen viability may be an important factor and selection criterion for cassava breeders.

Additionally, pollen grain viability was found to strongly and positively correlate with pollen grain size in strawberry, *Fragaria* × *ananassa* Duch [23], and *Mimulus guttatus* [24]. Relatedly, in *Lantana camara*, pollen with lower ploidy levels (diploids) was found to exhibit higher viability compared to that with higher ploidy levels (tetraploids, pentaploids, and hexaploids) [25]. Relatedly, high-ploidy-level amphiploid pollen grains in oats showed reduced viability compared with their parental species with lower ploidy levels [26].

Pollen viability can be estimated using a variety of methods, including in vivo and in vitro germination, as well as the staining techniques [17]. Another, more accurate method involves pollen deposition on stigma followed by evaluation of pollen tube growth and seed set [10,17]. In vitro techniques are mostly used because they are efficient, fast, and easy to assess. The chemical staining method is more advantageous as an indicator of pollen viability because it is faster and easier compared to pollen germination [10]. A more popular staining technique for assessing pollen viability involves the use of Alexander’s stain (AS), which excellently differentiates between aborted and non-aborted pollen [27]. Because different methods give different levels of accuracy in pollen viability evaluations, it is recommended that several test methods be used simultaneously. In this study, a combination of in vivo pollen germination and ovule fertilizability, stainability using AS, pollen grain diameter measurement, and ploidy level estimation were used to assess the viability of cassava pollen.

The question of whether pollen viability and its capacity to fertilize ovules is influenced by treating flowers with red light (RL) and plant growth regulators (PGRs) was the driving force behind this study. Thus, we aimed to assess pollen viability under these treatments using the in vivo pollen germination and seed set, pollen size measurement, and AS staining techniques, as well as to estimate the ploidy level using flow cytometry.

## 2. Results

### 2.1. Effects of RL and PGR Treatments on Pollen Characteristics and Their Capacity to Improve Fruit and Seed Set

Pollen characteristics, including pollen grain diameter and viability, measured as pollen stainability and capacity to fertilize ovules to cause fruit and seed set, were assessed on pollen samples picked from flowers of cassava plants subjected to extended daylength with RL and PGR treatments. Though there was no notable effect on the pollen diameter and its stainability, the treatment effect was significant (*p* ≤ 0.001) with respect to the capacity of pollen to fertilize ovules and cause fruit and seed set (Table 1). Treating flowers by exposing them to extended daylength or PGRs increased the capacity of the pollen to fertilize ovules (Figure 1), resulting in increased fruit and seed set efficiencies compared to the non-treated ones. Generally, the RL treatment was more effective in influencing pollen diameter, stainability, and capacity to cause fruit and seed set, except ovule fertilization.

### 2.2. Effect of RL and PGR Treatments on Pollen Stainability and Diameter

There was no statistical difference in the effects of PGR treatment on the pollen grain diameter and stainability compared to control (Table 2). However, stainability slightly increased, i.e., from 78.5 to 82.1%. Conversely, pollen grain diameter (148.5 µm) and stainability (93.0%) were significantly higher in flowers treated with RL compared to the non-treated ones (control), indicating that RL exposure may be involved in increasing the size and viability of pollen in cassava. Actually, pollen stainability in most flowers subjected to extended-daylength RL treatment was 100%.

More pollen grains were stained in flowers treated with RL (Figure 2(A1)) compared to PGR- and non-treated flowers (Figure 2(B1,C1)). Pollen grains that stained reddish purple were considered viable, while those stained pale green were considered non-viable. Additionally, some grains stained dark purple, implying higher viability, while others stained brightly red, implying a relatively lower viability. Pollen stainability was positively correlated with pollen diameter in both flowers subjected to treatment and those that were not, indicating that pollen viability is dependent on pollen size. The correlation was stronger (*R* = 0.61) in pollen from non-treated flowers (Figure 2(C2)), followed by that from PGR-treated flowers (*R* = 0.52) (Figure 2(A2)), and was weakest in RL-treated flowers (Figure 2(B2)). Overall, for all treatments, the correlation was *R* = 0.52.

### 2.3. Effect of Exposing Flowers to RL on Ovule Fertilizability and Fruit and Seed Set Efficiency

Exposing flowers to extended-daylength red light treatment significantly increased the capability of pollen to fertilize ovules (*p* ≤ 0.001), leading to enhanced fruit and seed set compared to control (Table 3). Efficiency of ovule fertilization, as well as fruit and seed set, did not vary significantly when either plants providing female or male flowers were treated with red light. However, the efficiencies varied significantly among the genotypes used in this study (Table 4).

In genotype UG15F180P005, treating the plants that provided male flowers with extended-daylength RL resulted in more ovule fertilization (53 vs. 25.4%), fruit set (59.5 vs. 53.0%), and seed set (39.4 vs. 30.1%) than when plants that provided female flowers were treated. Meanwhile, in UG15F302P016, ovule fertilization (62.2 vs. 56.9%), fruit set (64.8 vs. 60.1%), and seed set (40.5 vs. 38.4%) were slightly more efficient when plants providing female flowers were treated. Overall, subjecting both plants providing male and female flowers to red light treatment significantly increased pollen viability and thus increased ovule fertilization, and fruit and seed set efficiency.

### 2.4. Effect of Treating Flowers with PGR on Ovule Fertilizability and Fruit Set and Seed Set Efficiency

Treating female flowers with PGRs significantly increased the efficiency (*p* ≤ 0.001) of ovule fertilization and fruit and seed set compared to control (Table 5). In contrast, ovules fertilized when pollination was conducted using PGR-treated male flowers did not offer a significant difference from controls, although pollen from PGR-treated plants had higher fruit and seed set efficiencies (Table 5). The efficiency of seed set more than doubled under PGR treatment of either female or male flowers; whereas treating female flowers with BA resulted in the highest mean efficiency of ovule fertilization, the same treatment on male flowers caused their pollen to produce the lowest efficiency of ovules fertilization compared to control (Figure 3). Meanwhile, a combination of BA + STS caused the highest fruit and seed set efficiency.

Efficiency of ovule fertilization and fruit set and seed set varied among the study genotypes. In genotype UG15F192P012, the fruit set and seed set efficiencies were consistently higher in PGR-treated female flowers than the treated male flowers (Table 6). While in genotypes UG15F302P016 and UG15F222P017, the fruit set efficiencies were higher when pollen from PGR-treated flowers was used.

### 2.5. Comparison of Effects of RL and PGR Treatments on Pollen Viability and Fruit and Seed Set Efficiency

Exposing flowers to either RL or PGR treatments increased the efficiencies of ovule fertilization and fruit set and seed set compared to non-treated flowers (Figure 4). For treatments involving either female or male flowers, RL treatment improved fruit set and seed set efficiencies more than PGR treatment. In the case of the efficiency of the ovules fertilized, however, treating the female flowers with PGR was more effective than the treatment of male flowers (Figure 4A).

### 2.6. Pollen Ploidy and Its Relationship with Diameter and Viability in RL-Treated Flowers

Pollen from flowers exposed to red light treatment exhibited a slight increase in ploidy level compared to control (Figure 5A). Spearman’s correlation coefficients matrix was calculated for ploidy, pollen characteristics, and fruit and seed set efficiencies. Pollen ploidy level was positively correlated with pollen diameter (*R*^2^ = 0.09 *), ovule fertilization (*R*^2^ = 0.20 *), fruit set (*R*^2^ = 0.59 *), and seed set (*R*^2^ = 0.60 *) efficiency, but negatively correlated with pollen stainability (*R*^2^ = −0.37 *) (Figure 5B).

Variation in the amount of nuclear DNA and ploidy levels in pollen samples used in the flow cytometric analyses in this study is shown in Figure 6. The analyses showed that the DNA content in pollen grains ranged from *2C* (diploid) to polyploid amounts of *4C* (doubled), *6C*, and *8C* (more-than doubled), indicating increased ploidy levels due to endoreduplication of DNA. In determining the ploidy levels of the pollen samples, ploidy histograms were plotted and compared with the ploidy level of young cassava leaves (Figure 7). As shown in Figure 7A, the young leaf samples displayed one major *2C* peak, which clearly indicated a diploid state, and one minor *4C* peak, with the minor *4C* peak being most likely due to nuclei in the *G2* stage of the cell cycle. Comparatively, the histograms of the pollen samples predominantly displayed one ploidy peak within the *2C* DNA or diploid region, just as the control (Figure 7B–D) corresponded to the *G1* (DNA doubling) phase of the cell cycle. However, the peaks corresponded to a DNA content greater than 200, a channel mean set to correspond to *2n* DNA content of a diploid state. This shows that the pollen exhibited a slightly higher ploidy level. Taken together, the results indicate that the pollen grains picked at anthesis in this study contained diploid nuclei but had portions of higher ploidy nuclei.

## 3. Discussion

### 3.1. Effect of Extended Daylength with RL on Pollen Diameter, Stainability, and Ovule Fertilizability

The results of this study indicate that RL treatment exerted an effect on pollen viability by increasing pollen grain diameter, stainability, and capacity to fertilize ovules to cause fruit and seed set. The involvement of extended daylength with RL in the flowering process of crop plants has been extensively studied. It is involved in the regulation of various stages of plant development, including pollen development [28,29]. However, there are limited studies on the effect of RL on pollen viability or stainability in plants, including cassava.

In a previous study, RL increased pollen viability measured as in vitro pollen germination and pollen tube growth in *Pinus roxburghii* [30], while a mixture of red and blue light increased pollen viability in tomato [20]. These results closely agree with the findings of the current study, in which pollen viability measured as pollen stainability and ovule fertilizability were higher in flowers exposed to RL treatment. The physiological effect of RL in plants is linked to phytochromes, the light-sensitive pigments, existing in two interconvertible forms: Pr (an inactive, RL-absorbing) and Pfr (an active, far-red-light-absorbing) [31,32]. These interconversions have correlative photomorphogenic effects that result in flower formation [28]. The Pfr interacts with other proteins to trigger light-dependent photomorphogenic flowering responses [33,34]. It initiates signal transduction pathways within plant cells, which then influence gene expression and various physiological processes associated with floral initiation and development, including pollen development and viability [29]. Thus, the results in this study suggest that pollen stainability, in vivo germination, and ovule fertilization may also be phytochrome-controlled, and this is perhaps being reported for the first time. The observed variation in ovule fertilizability among genotypes could be attributed to physiological and/or genotypic differences in genes that are involved in the photoperiod and hormone systems that are involved in regulating flowering.

Pollen diameter was greater in RL-treated flowers compared to control. Though not directly comparable, positive correlation between pollen grain diameter and pollen viability was reported in sunflower [35] and *Mimulus guttatus* [24]. The larger pollen size could be attributed to evolutionary selection pressure imposed by pollinating honeybees, which prefer large-sized pollen grains [35,36], unreduced gametes [24], and a high protein content, mainly enzymes that function during pollen germination and subsequent fertilization [36]. Since exposure to extended daylength with RL influences Pr and Pfr interconversions, culminating in signaling which interacts with the circadian rhythm [37], it can be hypothesized that the increase in pollen diameter observed in this study might have been due to photoperiod signaling interactions.

### 3.2. Effect of PGR on Pollen Diameter, Stainability, and Ovule Fertilizability

In this study, the effects of two PGRs, BA and STS, on pollen diameter, stainability, and ovule fertilizability were investigated. The results indicate that there was no significant difference in pollen grain diameter and stainability between the pollen of PGR-treated and the non-treated flowers. However, PGR treatment increased stainability and the capability of the pollen to fertilize ovules, leading to increased fruit and seed set. The involvement of plant growth regulators (PGR) in modulating various morphogenetic processes, including flowering, has been reported [38,39]. However, the effect of PGR, especially BA and STS, on pollen viability or stainability and the capability of pollen to cause ovule fertilization in plants has not been widely explored. Moreover, the literature about this topic is scanty. Whereas BA was reported to induce the flowering or feminization of flowers, STS increased abundance and prevented abortion of flowers, thus enhancing fruit and seed set in cassava [22,40,41]. However, the effects of these PGRs on pollen viability and ovule fertilizability in cassava were not analyzed.

PGRs such as brassinosteroids were used to increase pollen viability and in vitro germination in male pomegranate flowers [16], while gibberellins were reported to be essential for pollen germination and pollen tube elongation in rice [42]. PGRs, especially cytokinins, promote protein synthesis and nutrient mobilization, cell division, and differentiation, leading to ovule and pollen development and viability, including pollen germination [43,44,45]. Application of BA, a cytokinin, slightly improved flowering and pollen viability in shallots (*Allium cepa* var. *ascalonicum*) [46]. On the contrary, in this study, a decrease in pollen viability and increased ovule fertilizability were observed when female and male flowers were treated with BA only compared to control. However, fruit and seed set increased compared to control.

Application of STS increased pollen quality in male flowers [47] and, contrary to the findings of this study, produced large-diameter pollen grains in both male and masculinized female genotypes of *Cannabis sativa* L. [48]. Relatedly, STS induced the production of male flowers with viable pollen grains, which were able to germinate on stigmas to effect seed set in genetically female *Cannabis sativa* plants [49,50]. All this can be explained by the fact that STS is known to prevent ethylene-induced senescence and to inhibit ethylene production by outcompeting copper ion cofactors in the ethylene receptors, allowing *CONSTITUTIVE TRIPLE RESPONSE1 (CTR1)* to be active and signaling a slowed ethylene response [51]. This, in turn, initiates male flower induction and maintains overall plant health and viability of flowers, including their pollen [52]. Similarly, the results of this study show increased pollen viability, exhibited through increased ovule fertilizability when both male and female flowers were treated with STS, and the consequent increase in fruit and seed set. These results confirm the effects of STS on pollen viability, thus corroborating the results of previous studies [47,50].

In the current study, significantly more promotive effects on the pollen’s capability to fertilize ovules and fruit and seed set were observed when BA and STS were applied in combination on the male and female flowers than when these growth regulators were applied singularly. This suggests that BA and STS had additive and/or synergistic or interactive effects, which increased pollen viability and ovule fertilizability. However, the variation in ovule fertilizability among genotypes could be attributed to physiological and/or genotypic differences in genes that are involved in the photoperiod and hormone systems that regulate flowering.

### 3.3. Relationship between Ploidy, Diameter, and Viability of Pollen in RL-Treated Flowers

Red light regulates developmental processes throughout plant life, including flower development [29,53]. The effect of RL on pollen ploidy has not been previously reported. However, in a previous study, extended photoperiod improved pollen quality in hexaploids better than in diploid and tetraploids and promoted the formation of apomictic seeds in the *Ranunculus auricomus* complex (Ranunculaceae) [54]. In this study, the pollen of flowers exposed to red light treatment exhibited a slightly increased ploidy level. Moreover, DNA content in the pollen grains of treated flowers ranged from a *2C* (diploid) amount to polyploid amounts of *4C* (tetraploid), *6C* (hexaploidy), and *8C* (octoploid)*,* indicating an increased amount of cellular DNA, and the ploidy peaks stabilized slightly beyond the control channel mean of 200. This suggests that red light may have played a role in stimulating increased DNA amounts. The increase in the DNA content indicates cell somatic polyploidization, which could be attributed to either endomitosis or endoreduplication. Endoreduplication, a more common mode of cell polyploidization in plants, involves one or several rounds of nuclear DNA synthesis in the absence of mitosis [55,56]. It occurs during biological processes, including cell metabolism, differentiation, and cell expansion. Endoduplication is presumably important for increasing the availability of DNA templates for gene expression [57].

Pollen size is often positively correlated with ploidy [58]; thus, is often used as a biological parameter by which to estimate the ploidy and viability of pollen [59]. It was previously reported that there is a tendency for polyploid pollen grains to be larger than diploid pollen grains [60,61]. In *Camellia oleifera*, pollen size was found to progressively increase with ploidy level, from diploids to octoploids, while pollen viability, as well as the germination rate, were relatively higher in hexaploids than in pollen with lower ploidy levels [59]. Similarly, in this study, pollen ploidy correlated positively with pollen diameter (Figure 5B). This is consistent with the results of a study in which it was found that the pollen grain size of octoploids was larger, thus suggesting that pollen size is influenced by ploidy levels [62]. Very little is known about the causal relationship between ploidy level and pollen size. In theory, polyploidy causes an increase in the amount of DNA and a consequent increase in the size of a cell [63]. Thus, the increased pollen diameter in relation to ploidy observed in this study could be due to an increased DNA amount or other genetic factors that may also play a significant role in determining pollen size.

Studies have shown that ploidy level affects pollen viability and pollen tube growth [60,64], with viability being higher in higher-ploidy pollens [61]. In this study, pollen ploidy correlated positively with ovule fertilization but negatively with pollen stainability. Partly in agreement with the findings of this study is that pollen stainability was higher in lower-ploidy pollen compared to higher-ploidy pollen in *Hydrangea macrophylla* [64] and *Lantana camara* [25], indicating a positive correlation. Conversely, the germination rate was higher in the high-ploidy hexaploid pollen than in the lower-ploidy tetraploid and diploid pollens in *Camellia oleifera*, indicating a negative correlation [59]. Although it has been reported that low ploidy levels confer lower pollen viability and stainability [61], in this study, it was not possible to establish a distinct relationship between ploidy level and the pollen viability and the stainability of pollen and its capacity to fertilize ovules. Thus, from the findings of this study, it has not been possible to clearly relate pollen ploidy level with pollen viability.

## 4. Materials and Methods

### 4.1. Plant Materials, Field Conditions and Treatments

This experiment was conducted using plant materials established under field conditions at the National Crops Resources Research Institute (NaCRRI), Namulonge, for two growing seasons (2019/2020 and 2020/2021, from June to June of each season), under rain-fed conditions. Namulonge is characterized by an average annual rainfall of approximately 1300 mm, an average annual temperature of 22 °C, and an annual minimum and maximum temperature of 16 and 28 °C, respectively. The data on average rainfall and temperature conditions that prevailed during this study (Appendix A) were obtained from the Namulonge meteorological station. The plant materials selected for this study included profusely or early to moderately flowering cassava genotypes. Three sets of plant materials (each consisting of 10 genotypes) were subjected to red light (RL-), plant growth regulators (PGR-), and non-treatment (Non), which acted as control (Appendix A).

The plant materials under extended daylength RL treatment were established in six experimental blocks. The RL treatment was administered as described in [22]. In this case, the study genotypes were subjected to a RL system setup using 50 W light emitting diode (LED) lamps with red LEDs (model 5-10x5w, Zhongshan, China) (illumination range 640–660 nm) with reflectors (339 × 350 mm, model ISL-RFGB, CCS Inc., Zhongshan, China) as sources of RL for the extension of the photoperiod during the night. In each block, a lamp was placed horizontally at 3 m above the ground in the center of the block to cast red light over the plants up to a radius of 6 m, with an intensity ranging from 1.0 to ≤0.5 PFD (photo flux density of wavelengths: 400–700 nm) in μmol m^−2^ s^−1^. Plants beyond this radius were in a region of no light, i.e., total darkness (0PFD in μmol m^−2^ s^−1^), and thus served as control. The light intensity was measured with a Licor quantum sensor (model LI-190; Lincoln, Nebraska, NE, USA). The illumination was commenced soon after complete sprouting/germination (about 14 days after planting) and then administered daily throughout the night, from dusk to dawn, until the plants reached 2nd to 4th level of flowering.

The PGR-treated plants were established in a randomized split-plot design, in which each variety was represented by 20 plants, spaced at 1 m between plants and rows, in three blocks or replications. Each main plot (genotype) was split into four plots of four plants each. Two PGRs, silver thiosulphate (STS) and 6-benzyl adenine (BA), which came through as the most effective candidates following the screening of several PGRs for their effect on cassava flowering [65,66], were used in this study. These PGRs were optimized in a study by [40]. The PGRs were applied in four treatment regimens: (1) BA only; (2) STS only; (3) combination of BA and STS; and (4) control (non-treatment). One treatment was administered per split plot, following a method described by [22]. Briefly, 2.5 mL of 4 mM STS were applied through the petiole at a 14-day interval, while 0.5 mM BA was applied via hand spray at shoot tips until just wet at a 7-day interval. The STS solution was prepared following a modification of the method previously described and optimized by [41]. In this case, 1 part of 0.1 M silver nitrate (AgNO_3_) (Sigma-Aldrich, Massachusetts, USA) was added drop-wise to four parts of 0.1 M sodium thiosulfate (Na_2_S_2_O_3_) (Sigma-Aldrich, USA) and diluted with distilled water to the desired concentrations and volumes. The BA solution was prepared by diluting a 6.38 mL (*v*/*v*) BA (Sigma-Aldrich, USA or Duchefa Biochemie, Haarlem, The Netherlands) stock (1.765 g/100 mL) with distilled water to 1 L of solution. The treatments commenced at the earliest notice of forking in any one genotype, and this was routinely continued up to the fourth tier/level of branching, 5 to 8 months after planting, as this varied with genotype.

### 4.2. Pollen Collection

The male or staminate flowers, as sources of pollen, were picked in batches from each of the RL-, PGR-, and non-treated plants, and each batch was assessed for viability using the techniques of in vivo germinability and capacity to set fruit and seed, in vitro staining, and measurement of grain diameter and ploidy level. The flowers were sampled from forking/branching levels between one and four. In each case, mature and ripe male flower buds (anthers) were hand-picked in the mornings before midday. To allow for replication, batches of pollen samples were picked at least two times from a particular flowering level.

### 4.3. Assessment of Pollen Germination, Stainability, and Measurement of Grain Diameter

#### 4.3.1. Pollination

To assess the ability of RL-, PGR-, and non-treated (control) pollen to germinate on stigmas and its capacity to effect seed set (ovule fertilizability), self-, out-, and hybridization crosses were conducted through manual pollinations. In this case, mature female flowers on plants under RL, PGR, and control treatments were bagged using muslin bags for 1 to 3 days prior to anthesis to avoid contamination by pollen from unknown sources. Pollen samples collected in Section 4.2 from same and different genotypes were used to pollinate fully opened flowers (at anthesis) by gently brushing the opened anthers on the stigma. All the pollinated flowers were re-bagged for at least three days to keep away stray pollen.

#### 4.3.2. In Vivo Pollen Germination and Ovule Fertilization

To determine the germinability of the pollen and ovule fertilization, about three hand-pollinated flowers were picked at two-to-three days after pollination (DAP), fixated in a solution containing glacial acetic acid and 96% ethanol (in a ratio of 1:3), and then stained with 0.1% aniline blue in 0.1% K_2_HPO_4_ before being examined for pollen tube growth, as previously described in [67]. The pistils were removed from the stain and placed in a drop of glycerol on a glass slide; the stigmas cut off and the ovaries were dissected to extract ovules. Ovary wall tissues were discarded; a drop of basic 0.1% aniline blue solution was added on the ovules, covered with a cover slip and then gently squashed. Observations of pollen germination and pollen tube growth were made with a fluorescence microscope (Nikon Alphaphot-2 YS2, Nikon Corporation, Tokyo, Japan), and images were taken using a camera head (Nikon DS-L3, Nikon Corporation, Tokyo, Japan).

#### 4.3.3. In Vitro Pollen Stainability

In the in vitro viability test using the staining assay, a portion of the fresh male flowers were fixated in a Carnoy solution (containing absolute ethanol, chloroform, and glacial acetic acid in a ratio of 6:3:1, respectively) for no less than 2 h. Stainability was determined by using a modified solution of Alexander Stain prepared according to a protocol described by [27]. The final stain solution, which was kept in the dark, contained the following constituents, added in the order they are presented: 10 mL 95% alcohol; 1 mL malachite green (1% solution in 95% alcohol); 50 mL distilled water; 25 mL glycerol; 5 mL acid fuchsin (1% solution in water); 0.5 mL orange G (1% solution in water); 4 mL Glacial acetic acid; and distilled water (4.5 mL), added to make a total volume of 100 mL solution.

The flowers were carefully removed from the fixative solution and placed on absorbent paper to dry off excess solution. Two-to-three buds were then placed on the glass slide, dissected under a dissecting microscope to release the pollen, and the leftover plant debris were carefully removed. Two-to-four drops of the Alexander stain were added before the sample completely dried, and the slide was slowly heated over a spirit burner in a fume hood until the stain solution was near boiling (~10 s). This was undertaken to enable better penetration of the dye into the cellulose and protoplasm of the pollen. The sample was left to stand for 10 to 15 min after heating to allow the stain to be completely absorbed into the pollen grains. A cover-slip was placed over the sample, and even pressure was applied on the cover-slip to ensure that the pollen converged on one plane. The slides were then examined using a microscope (Nikon, Model C—PS 1092883, Nikon Corporation, Tokyo, Japan), and images were taken using a camera head (Nikon DS-L3, Japan). Pollen grains that stained red to purple were regarded as viable, while those that stained green were non-viable. Viable and non-viable pollen were manually quantified by counting differentially stained pollen in three views in three replications (slides). Percentage (%) pollen stainability per view was calculated as follows:$$Pollen\;stainability\left(\%\right)=\frac{Stained\;pollen\;grains\;in\;a\;view}{Total\;pollen\;grains\;in\;a\;view}\times 100.$$

#### 4.3.4. Estimation of Pollen Grain Diameter

To estimate pollen size, the temporary slides prepared for the in vitro stainability tests above were first used to measure grain diameter because fixation did not affect grain size when compared with the fresh pollen. The choice to measure diameter was based on the fact that cassava pollen is basically spherical in shape. In this case, 20 pollen grains were randomly sampled per slide, and their diameters were measured under an ocular microscope (Optika, SZM-1LED, Ponteranica, Italy) using a calibrated eyepiece lens at a magnification of 10×. For the purposes of replication, sampling and measurement were conducted three times per slide.

### 4.4. Assessment of Fruit and Seed Set Efficiency

To determine fruit and seed set following controlled pollination, the pollinated flowers were observed and monitored for fruit set until harvest. At two-to-three months after pollination, the fruits were harvested to determine seed set. Fruit (FE) and seed efficiencies (SE) were calculated in percentages as follows:$$Fruit\;set\;efficiency\left(FE\right)\;(\%)=\frac{No\;of\;fruits\;set}{No\;of\;fruits\;expected}\times 100,$$
$$Seed\;efficiency\left(SE\right)\;(\%)=\frac{No\;of\;seeds\;set}{No\;of\;seeds\;expected}\times 100.$$

### 4.5. Determination of Pollen Ploidy Level

Pollen samples picked in Section 4.2 were used to measure ploidy levels. Since the samples were collected in small bits, they were dried in silica gel and kept until a sizeable amount for analysis was obtained. The dry samples were kept (on silica gel) at room temperature for no more than four months. The flow cytometry method was used to determine the amount of DNA and the ploidy level of the preserved pollen following a modification of the method described by [68]. In brief, two-to-five male flower buds with dehisced anthers were placed in a 2 mL Eppendorf tube, 1 mL of cold Otto1 buffer (0.1 M citric acid monohydrate and 0.5% *v*/*v* of Tween-20) was added, and they were allowed to soak for five minutes then vortexed slightly for a few seconds to shed off the pollen. The pollen suspension was placed on a clean glass petri dish, and the pollen grains were gently crushed against the glass for 10–15 s using a glass rod with a flattened end. Then, the homogenate was filtered through a 50 µm nylon filter into a cuvette. The samples were incubated for about 20–60 min before 1 mL of Otto II buffer (0.4 M anhydrous Na_2_HPO_4_, 4 µg/mL of DAPI (4,6′-diamidino-2-phenylindole), and 1 µL/mL β-metcaptoethanol) were added and ran according to standard plant flow cytometry protocols using a Sysmex Partec Ploidy Analyser Machine (the CyFlow^®^ ploidy analyser, Software—CyViewTM 1.6, REF-CY-S-3039, Münster, Westf, Germany). For control purposes, young leaves from one test plant were used. To extract nuclei and DNA, the leaves were chopped with a sharp razor blade in a cold buffer and then taken through the procedure for ploidy determination as described above. The *G1* (*2n* DNA content) peak of the control was set at channel 200. The ploidy level of the pollen sample was determined by comparing the relative position of its *G1* peak and that of the control. Data analyses were controlled and performed within the ploidy analyzer by the CyView™ software (CyViewTM 1.6).

### 4.6. Statistical Analysis

All statistical analyses were undertaken using statistical packages in R (v.4.2.2) [69]. A generalized linear mixed-effects model (GLMM) was used to investigate the determinants of the number of fertilized ovules, stained pollen grains, and fruit and seed set, assuming Poisson distribution. Data on pollen diameter were assumed to be continuous and thus were analyzed using one-way analysis of variance (ANOVA). The graphics were prepared using the Grammar of Graphics (ggplot2) package. The Spearman coefficient of rank correlation was used to examine relationships among traits. Where applicable, the means were compared using Tukey’s grouping test, with a 5% probability. Percentage pollen stainability and fruit and seed set efficiencies were computed using the expressions shown in Section 4.3.3 and Section 4.3.4, respectively.

## 5. Conclusions

Enhancing pollen viability boosts crossing programmes in cassava breeding. Application of RL and PGRs (BA and STS) improved pollen viability and thus fruit and seed set. Red light treatment increased pollen diameter, stainability, ovule fertilizability, and ploidy level. PGR treatment did not affect pollen diameter and stainability but increased ovule fertilizability. Further studies are necessary to clarify the influence of RL and PGR (BA and STS) on pollen viability.

## Figures and Tables

**Figure 1 plants-13-01988-f001:**
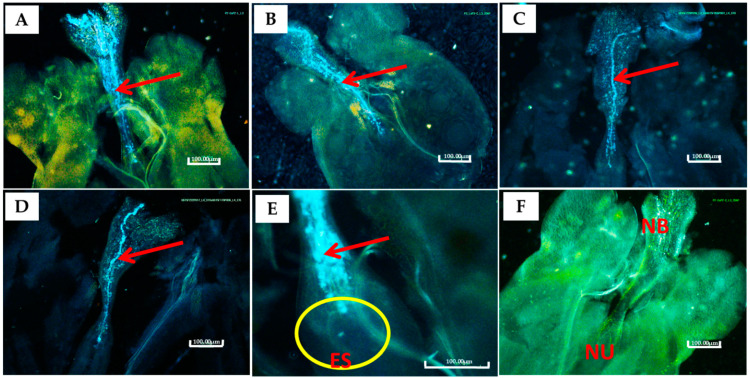
PGR- or RL-treated flowers, pollen germination, and ovule fertilization: (**A**) ovule of untreated female flower x pollen of RL-treated male flower; (**B**) ovule of RL-treated female flower x pollen of untreated male flower; (**C**) ovule of untreated female flower x pollen of PGR-treated male flower; (**D**) PGR-treated female flower x pollen from untreated male flower; (**E**) pollen penetration into embryo sac (**ES**) and fertilization; (**F**) ovule of untreated female flower x pollen of untreated male flower. **NB**—nucellar beak; **NU**—nucellus. Red arrows point at pollen tubes, and a yellow circle surrounds an embryo sac region within the nucellus. Scale bar = 100 μm.

**Figure 2 plants-13-01988-f002:**
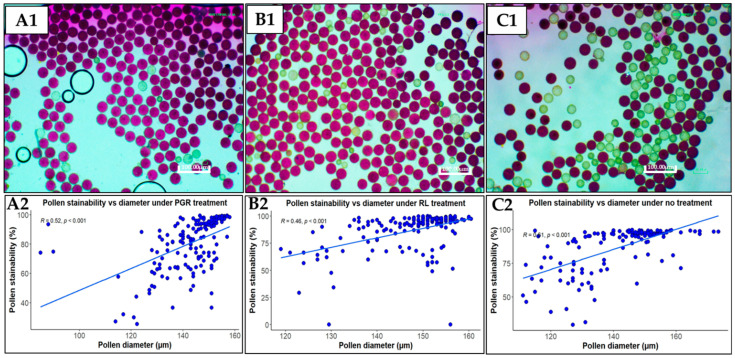
Pollen stainability and its correlation with diameter in flowers subjected to different treatments: (**A**) plant growth regulators (PGR) treatment; (**B**) red light (RL) treatment; (**C**) no treatment. (**A1**–**C1**) show pollen stained with Alexander staining and observed at 10× magnification using an inverted microscope and images taken with a camera head (Nikon DS-L3, Nikon Corporation, Tokyo, Japan). Pollen grains stained reddish purple are viable, while those stained pale green are unviable. Scale bar = 100 μm. (**A2**–**C2**) show correlations between stainability and diameter of pollen under PGR, RL and no treatments respectively.

**Figure 3 plants-13-01988-f003:**
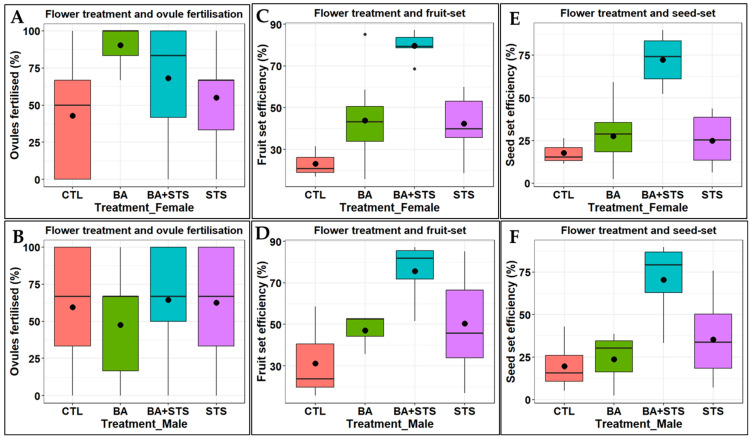
Mean efficiency of ovule fertilization and fruit and seed set in flowers subjected to PGR treatments: (**A**) ovules fertilized in treated female flowers; (**B**) ovules fertilized using pollen from treated flowers; (**C**,**D**) fruit set when female and male flowers were treated, respectively; (**E**,**F**) seed set when female and male flowers were treated, respectively. The black lines inside each boxplot represent the median value, and the dots inside each boxplot represent the mean value.

**Figure 4 plants-13-01988-f004:**
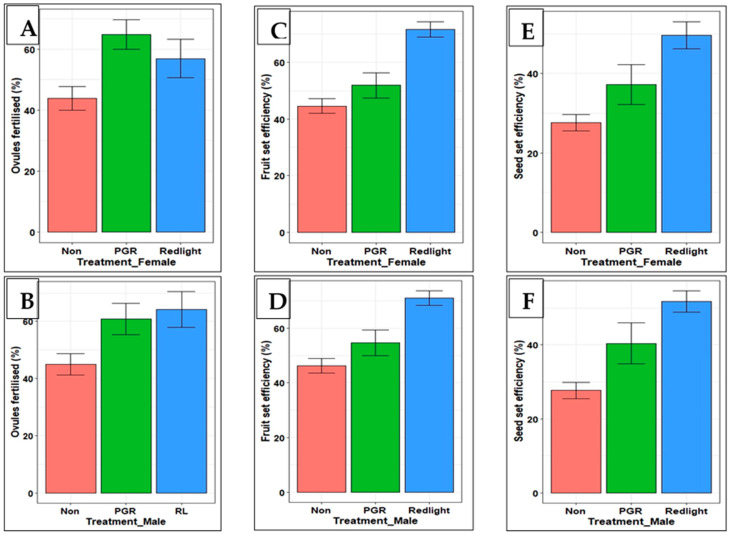
Comparison of effects of subjecting flowers to PGR and RL treatments on the efficiency of ovule fertilization and fruit and seed set: (**A**) ovules fertilized in treated female flowers; (**B**) ovules fertilized using pollen from treated male flowers; (**C**,**D**) fruit set when female and male flowers were treated, respectively; (**E**,**F**) seed set when female and male flowers were treated, respectively. Shown are means of replicates across flowering levels 1 to 4 and two growing seasons; 95% confidence intervals are indicated by standard error bars (ovules fertilised, *n* = 223; fruit set, *n* = 115; seed set, *n* = 115).

**Figure 5 plants-13-01988-f005:**
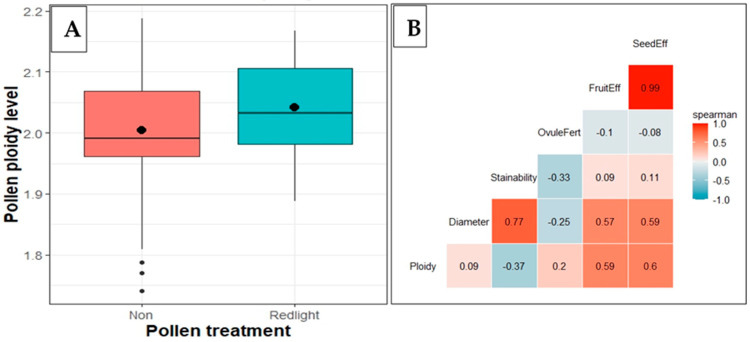
Cassava pollen ploidy and correlations: (**A**) effect of RL treatment on ploidy level (the dots inside each boxplot represent the mean value); (**B**) correlation between ploidy and pollen characteristics.

**Figure 6 plants-13-01988-f006:**
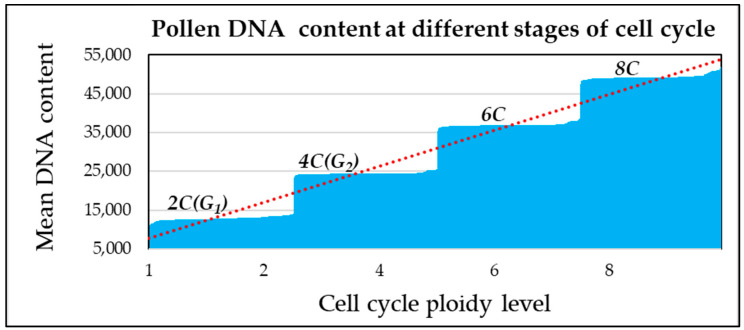
Flow cytometry profile for cassava pollen samples analyzed for DNA content and ploidy levels: *2C* represents the diploid amount of DNA at the G1 stage of the cell cycle, while the *4C* is the doubled amount at the G2 stage following DNA synthesis; *6C* and *8C* show DNA amounts due to endoreduplication.

**Figure 7 plants-13-01988-f007:**
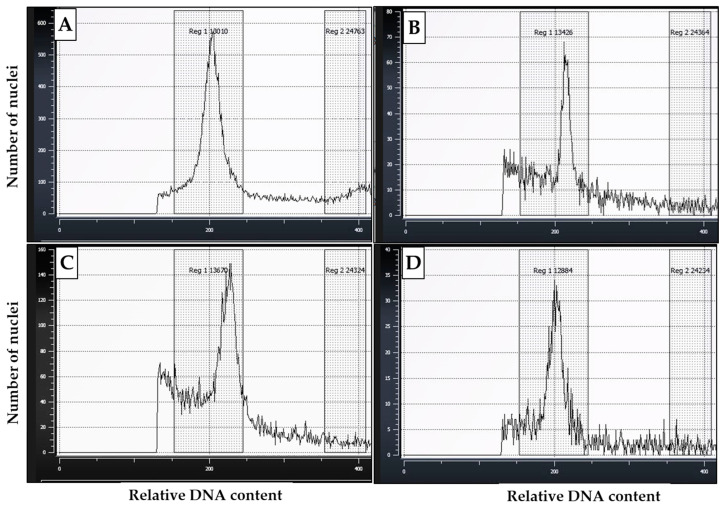
Representative flow cytometric histograms of pollen and leaf nuclear preparations for cassava: (**A**) cassava young leaf (control); (**B**,**C**) pollen samples picked at anthesis from RL-treated flowers; (**D**) pollen sample picked at anthesis from non-treated plants.

**Table 1 plants-13-01988-t001:** Variation of pollen characteristics and its ability to cause fruit and seed set when treated with RL and PGR.

Variable	Pollen Diameter (µm)			Pollen Stainability (%)			Ovules Fertilized (%)			Fruit Set Efficiency (%)			Seed Set Efficiency (%)		
Treatment	RL	PGR	Ctl	RL	PGR	Ctl	RL	PGR	Ctl	RL	PGR	Ctl	RL	PGR	Ctl
Mean	148.5	142.4	145.6	93.0	82.1	78.5	57.0	65.0	44.4	71.8	51.9	44.6	49.7	37.3	27.7
SEM	0.7	1.0	1.2	1.4	1.4	1.5	4.2	4.5	6.0	2.2	4.3	4.2	2.3	4.7	3.0
CI	1.4	1.9	2.4	2.7	2.8	2.9	8.3	8.9	12.2	4.4	8.8	9.2	4.6	9.5	6.7
Variance	83.5	150.7	191.9	305.5	318.7	284.3	1856.2	1381.9	1714.9	368.3	511.0	209.9	393.2	606.0	111.0
Std	9.1	12.3	13.9	17.5	17.9	16.9	43.1	37.2	41.4	19.2	22.6	14.5	19.8	24.6	10.5
CV	0.1	0.1	0.1	0.2	0.2	0.2	0.8	0.6	1.2	0.3	0.5	0.5	0.5	0.7	0.6
Significance (α = 0.05)	ns	***		***	ns		ns	***		***	***		***	***	

RL = red light; PGR = plant growth regulator; Ctl = control; SEM = standard error of mean; CI = confidence interval; Std = standard deviation; CV = coefficient of variation. Tests of significant treatment effects are indicated: *p* ≤ 0.001 (***), ns—not significant.

**Table 2 plants-13-01988-t002:** Pollen diameter and stainability under PGR and RL treatments.

Treatment	Pollen Diameter (µm)	Pollen Stainability (%)
PGR	142.4 b	82.1 b
Red light	148.5 a	93.0 a
Control (Non)	145.6 b	78.5 b

Different lower-case letters within columns indicate significant differences among treatments via Tukey’s HSD test (*p* ≤ 0.05).

**Table 3 plants-13-01988-t003:** Efficiency of ovule fertilization and fruit and seed set in flowers subjected to RL treatment.

Treatment	Ovules Fertilised (%)		Fruit Set Efficiency (%)		Seed Set Efficiency (%)	
	Female	Male	Female	Male	Female	Male
Red light	56.9 a	64.3 a	71.8 a	71.1 a	49.7 a	51.8 a
Control	52.0 b	47.7 b	51.5 b	53.7 b	31.7 b	31.9 b
Significance (α = 0.05)	***	***	***	***	***	***

Different lower-case letters within columns indicate significant differences among treatments via Tukey’s HSD test (*p* ≤ 0.05). ANOVA tests of significant main effects are indicated: *p* ≤ 0.001 (***).

**Table 4 plants-13-01988-t004:** Efficiency of ovule fertilization and fruit and seed set in flowers of cassava genotypes subjected to extended-daylength RL treatment.

Genotype	Ovules Fertilised (%)		Fruit Set Efficiency (%)		Seed Set Efficiency (%)	
	Female	Male	Female	Male	Female	Male
UG15F178P001	65.1 a	66.7 a	54.7 b	58.1 c	37.0 c	41.2 a
UG15F180P005	25.4 c	53.0 b	53.8 b	59.5 bc	30.1 d	39.4 ab
UG15F192P012	57.1 b	43.3 c	63.9 a	65.9 a	49.6 a	40.7 a
UG15F302P016	62.2 a	56.9 b	64.8 a	60.1 b	40.5 b	38.4 b
Significance (α = 0.05)	ns	***	***	***	***	***

Different lower-case letters within columns indicate significant differences among genotypes via Tukey’s HSD test (*p* ≤ 0.05). ANOVA tests of significant main effects are indicated: *p* ≤ 0.001 (***); ns—not significant.

**Table 5 plants-13-01988-t005:** Ovule fertilization and fruit and seed set in female and male flowers subjected to BA + STS PGR treatment.

Treatment	Ovules Fertilized (%)		Fruit Set Efficiency (%)		Seed Set Efficiency (%)	
	Female	Male	Female	Male	Female	Male
PGR	64.9 a	60.9 a	51.9 a	54.7 a	37.3 a	40.4 a
Control	42.9 b	59.4 a	23.2 b	31.3 b	17.7 b	19.6 b
Significance (α = 0.05)	***	ns	***	***	***	***

Different lower-case letters within columns indicate significant differences among treatments via Tukey’s HSD test (*p* ≤ 0.05). ANOVA tests of significant main effects are indicated: *p* ≤ 0.001 (***), ns: not significant.

**Table 6 plants-13-01988-t006:** Efficiency of ovule fertilization and fruit and seed set in flowers of cassava genotypes treated with PGR.

Genotype	Ovules Fertilized (%)		Fruit Set Efficiency (%)		Seed Set Efficiency (%)	
	Female	Male	Female	Male	Female	Male
NASE14	-	90.0 a	49.5 b	55.7 b	30.7 bc	42.0 ab
UG15F039P015	-	46.7 de	-	68.6 a	-	52.3 a
UG15F056P001	-	79.2 b	79.5 a	35.9 e	57.3 a	30.2 cde
UG15F192P012	81.5 a	33.4 e	72.2 a	41.8 cde	68.4 a	25.1 de
UG15F222P017	48.7 c	-	32.8 d	58.6 ab	25.0 c	26.4 de
UG15F228P016	57.5 b	-	40.4 c	-	23.8 c	-
UG15F302P016	83.3 a	41.7 e	49.5 b	63.1 a	36.0 b	43.7 ab
Significance (α = 0.05)	*	ns	*	***	***	***

Different lower-case letters within columns indicate significant differences among genotypes via Tukey’s HSD test (*p* ≤ 0.05). ANOVA tests of significant main effects are indicated: *p* ≤ 0.05 (*), *p* ≤ 0.001 (***); ns: not significant.

## Data Availability

All relevant datasets generated during and/or analyzed during the current study can be found in online repositories and are available from Cassavabase, https://www.cassavabase.org/breeders/trial/6707?format= (accessed on 7 June 2024), a website maintained by the Next Generation Cassava Breeding Project.

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
