# Peer review of "Assessment of Cassava Pollen Viability and Ovule Fertilizability under Red-Light, 6-Benzyl Adenine, and Silver Thiosulphate Treatments"

_plants, 2024, doi:10.3390/plants13141988_

Round 1

Reviewer 1 Report

Comments and Suggestions for Authors

Summary Comments:

The paper compares two treatments RL and PGR in cassava, which show the pollen viability and ovule fertilizability increased, also the fruit and seed set efficiency higher. These treatments will help cassava to give more seeds to increase the plant viability. But still have some points that are not so clear.

Areas for improvement:

1> The results in the abstract are too detailed and the authors should summarise the main findings and conclusions.

2> For the two treatments, how do we know the optimal time for RL or PGR treatment and the optimal concentration of PGR?

3> For the results in Figure 1, Table 1 shows that pollen viability or diameter increased only slightly with RL or PGR treatments, so the control stigma should have some germinated pollen. The authors should have made more than 10 stigmas to count successfully fertilized pollen.

4> It would be better to show pollen germination in vitro to prove that the dark purple pollen grains are more viable. Based on the results of Figure 2.

5> How do you explain the different fertilization and fruiting rates of the ovules but similar seed setting rates for different genotypes?

6> How can these two treatments be carried out simultaneously or sequentially on cassava? Because of the change in pollen size after the RL treatment, is it likely that the fruiting rate will increase with a further PGR treatment?

7> What about seed quality after treatment during cassava growth, when pollen ploidy is increased? Or does the whole plant become stronger in the next generation?

Reviewer 2 Report

Comments and Suggestions for Authors

Cassava has a high rate of flower or fruit abortion. Previous studies found that red light or plant growth regulator treatment could significantly improve fruit and seed set. On this basis, the manuscript by Baguma et al further analyzed the effects of the above treatments on pollen viability, and found that red light treatment increased pollen diameter, stainability and ovule fertilizability, and PGR treatment also increased ovule fertilizability, thus positively affecting fruit and seed set. The study explains, to a certain extent, why treatments such as red light can improve fruit set, providing reference for cassava breeding and seed production. The description of the experimental design and measurement methods are clear. The interpretation of the data the conclusion is appropriate. However, my understanding of the data analysis process is not very clear:

1, In Table 1, are the RL, PGR, and Non data based on same genotypes? Is it calculated using data from all 10 genotypes tested? Is PGR a collection of BA, STS, BA+STS related data? There are corresponding controls for RL and corresponding controls for PGR. Then Non is?

2, In Table 2, mean pollen stainability under RL treatment was 93%, while in Table 1 it was 86.6%. Why are these two different? What is the sample size respectively?

3, Line 169. “Actually, pollen stainability ranged from 0% in non-treated flowers to 100% in flowers treated with RL.” Please explain this sentence.

4, In Table 4, how do these genotypes perform when neither male nor female flowers are treated with RL?

5, In table 6, lower-case letters within male columns might be incorrect.

Reviewer 3 Report

Comments and Suggestions for Authors

Major:

1) Authors should improve all used figures in manuscript (Ms) text, e.g.

a) Fig. 1 and 7: increase figure quality;

b) Fig. 2, 3, and 7: enhance the font size.

c) Fig. 1, and 2: include the bar (mm).

d) Authors should improve used legends for figures, e.g. explain all used abbreviations, statistical treatment, etc.

e) Authors should include the statistical treatment in the Fig. 4.

It is hard to understand the results of the Ms without these corrections.

2) This Ms contains misprints, mistakes in English grammar and in the writing style. I recommend that the authors should use some help of a native English speaker or send the Ms to an English Editing Service that proofreads scientific writing.

3) Authors should explain used concentration of cytokinin (6-benzyl adenine, BA) and anti‑ethylene (silver thiosulphate, STS) and durations of the red light. Why these treatments? Why only one dose and one treatment time? It is usually advisable to use a minimum of 2 doses.

4) The authors should describe in more detail the procedure for plants treatment (by BA, STS, and red light), it may even be worth presenting it in the form of a schematic figure, in the form of supplementary material.

5) Could the authors explain whether any other parameters were measured in the studied cassava Manihot esculenta plants? For example, the size of tubers, because it is an important economic indicator for this plant.

Comments on the Quality of English Language

Moderate editing of English language required

Reviewer 4 Report

Comments and Suggestions for Authors

Climate change and the progressive increase in population make it increasingly necessary to research on improving the production of crops that can help human nutrition. That's why I think this article is interesting and goes along that line of research.

The conclusions are in accordance with the objectives.

Only some acronyms appear in the legend of Table 1. They must all appear, so that the table is better understood.

In the legend of Figure 2, is the last sentence correct?: “Pollen grains stained reddish purple are viable while those stained pale”

The legends of Figures 6 and 7 are partly overlapping and are not well understood.

On line 586 “Pollen samples collected in section 2.2” shouldn't I put “Pollen samples collected in section 4.2”?

And the same thing happens on line 646: “Pollen samples picked in section 2.2 were used to measure ploidy levels” should be “section 4.2”

In the “References” part, after the authors and before the title of the articles, I think they should be separated by a period and not by a comma.

Reviewer 5 Report

Comments and Suggestions for Authors

In this manuscript (plants-3072075) entitled "Assessment of Cassava Pollen Viability and Ovule Fertilizability under Red-Light and Plant-Growth-Regulator Treatments" submitted to Plants, Julius K. Baguma and colleagues have iinvestigated effects of field-applied red light (RL) and plant growth regulators (PGRs) on pollen viability and ovule fertilizability. This research is interesting and convincing, but this present manuscript needs revisions before publication.

Major points:

1. For Figures 1 and 2, scale bar should be included in the revised microscopy figures

2. At least three different concentrations of plant growth regulators (PGRs) should be examined for their effects on pollen viability and ovule fertilizability in the revision.

3. At least three different intensities and/or exposure times of red light (RL) should be examined for their effects on pollen viability and ovule fertilizability in the revision.

4. For Figures 3, 4, 5, and 6, at least three repeats should be conducted, and analysis in significance of difference should be performed. Please exhibit the significance of difference in the revised Table 1

Minor points:

1. Authors need to standardize references according to the Plants template. For instance, abbreviation instead of full name of Frontiers in Sustainable Food Systems should be presented (Reference 3).

Round 2

Reviewer 1 Report

Comments and Suggestions for Authors

Most of my concerns have been satisfactorily answered.

Author Response

Comment: Are the methods adequately described? Must be improved

Response:  More details have been included in the methodology, including a graph of rainfall and temperature changes during the growing seasons as Supplementary figure 1. These can be checked out in Section 4.1, lines 555-60; 565-578; 590-596 and Supplementary figure 1

Reviewer 3 Report

Comments and Suggestions for Authors

Major:

Just one significant point:

a) The authors should describe in more detail the procedure for plants treatment (by BA, STS, and red light):

the growing season, irrigation features, lighting, red light treatment time, the volume/concentrations/treatment methods of growth regulators in plots, etc.

b) Include all the main features of plant growth and treatment in the Supplementary

Figure 1.

Minor:

In Title: “and Plant-Growth-Regulator Treatments” correct to “6-Benzyl Adenine and Silver Thiosulphate Treatments”.

Comments on the Quality of English Language

Moderate editing of English language required

Author Response

Comment 1: The authors should describe in more detail the procedure for plants treatment (by BA, STS, and red light): the growing season, irrigation features, lighting, red light treatment time, the volume/concentrations/treatment methods of growth regulators in plots, etc.

Response:  Details of plant treatments have been included in the methodology.  A graph of rainfall and temperature changes during the growing seasons has also been included as Supplementary figure 1. These can be checked out in Section 4.1, lines 555-60; 565-578; 590-596 and Supplementary figure 1

Comment 2: Include all the main features of plant growth and treatment in the Supplementary Figure 1.

Response: Additional main features have been added to the supplementary figure. Please refer to Supplementary figure 2

Comment 3: In Title: “and Plant-Growth-Regulator Treatments” correct to “6-Benzyl Adenine and Silver Thiosulphate Treatments”.

Response: The title has been corrected as suggested. it now reads: "Assessment of Cassava Pollen Viability and Ovule Fertilizability under Red-Light, 6-Benzyl Adenine and Silver Thiosulphate Treatments"

Comment 4: Moderate editing of English language required

Response: An attempt to edit the English language has been made